# Identifying Salt Marsh Shorelines from Remotely Sensed Elevation Data and Imagery

**Amy S. Farris** *[ID], **Zafer Defne**[ID] **and Neil K. Ganju**

Woods Hole Coastal and Marine Science Center, U.S. Geological Survey, Woods Hole, MA 02543, USA
* Correspondence: afarris@usgs.gov; Tel.: +1-508-457-2344

**Abstract:** Salt marshes are valuable ecosystems that are vulnerable to lateral erosion, submergence, and internal disintegration due to sea level rise, storms, and sediment deficits. Because many salt marshes are losing area in response to these factors, it is important to monitor their lateral extent at high resolution over multiple timescales. In this study we describe two methods to calculate the location of the salt marsh shoreline. The marsh edge from elevation data (MEED) method uses remotely sensed elevation data to calculate an objective proxy for the shoreline of a salt marsh. This proxy is the abrupt change in elevation that usually characterizes the seaward edge of a salt marsh, designated the "marsh scarp." It is detected as the maximum slope along a cross-shore transect between mean high water and mean tide level. The method was tested using lidar topobathymetric and photogrammetric elevation data from Massachusetts, USA. The other method to calculate the salt marsh shoreline is the marsh edge by image processing (MEIP) method which finds the unvegetated/vegetated line. This method applies image classification techniques to multispectral imagery and elevation datasets for edge detection. The method was tested using aerial imagery and coastal elevation data from the Plum Island Estuary in Massachusetts, USA. Both methods calculate a line that closely follows the edge of vegetation seen in imagery. The two methods were compared to each other using high resolution unmanned aircraft systems (UAS) data, and to a heads-up digitized shoreline. The root-mean-square deviation was 0.6 meters between the two methods, and less than 0.43 meters from the digitized shoreline. The MEIP method was also applied to a lower resolution dataset to investigate the effect of horizontal resolution on the results. Both methods provide an accurate, efficient, and objective way to track salt marsh shorelines with spatially intensive data over large spatial scales, which is necessary to evaluate geomorphic change and wetland vulnerability.

**Keywords:** marsh edge; marsh shoreline; unmanned aircraft system; UAS; UAV; drone; lidar; salt marsh; coastal wetlands; Plum Island

## 1. Introduction

Salt marshes are ecologically important and valuable coastal features [1,2]. They provide critical habitat for many species, decrease storm surge by attenuating waves, improve ecosystem health by cycling nutrients and they sequester carbon [3–5]. Unfortunately, salt marshes are increasingly threatened by anthropogenic factors [6–9], as well as sea level rise [10,11] and sediment deficits [12]. Marsh health can also be influenced by plant–herbivore interactions [13,14]. Given the substantial ecosystem services provided by marshes, it is important to be able to measure their extent and seaward edge at high temporal and spatial resolutions. Within individual marsh complexes, progress has been made in identifying unvegetated and vegetated areas in order to compute actual marsh coverage, using remote sensing [11,12,15], however, tracking the seaward edge along salt marsh shorelines is less evolved. Although many studies have suggested ways to monitor salt marsh extent, it can be valuable

to have multiple techniques, so that scientists can choose the best method based on data availability and their application.

A common method for calculating a salt marsh shoreline is to digitize the visually identifiable edge of vegetation from visible band ortho-photos [16–18]. However, this method is relatively labor intensive and difficult to reproduce because different operators may digitize the shoreline differently.

There have been many studies that investigate marsh shorelines. The following list is intended to be illustrative, not exhaustive. The Coastal Mapping Program (CMP) (https://www.ngs.noaa.gov/RSD/cmp.shtml) run by the National Oceanic and Atmospheric Administration's (NOAA) National Geodetic Survey (NGS) has well-developed procedures for mapping salt marsh shorelines [19–21]. Campbell and Wang [22] used imagery analysis to study the extent of marsh grass, and the evolution of pannes and pools. Kulei, Guneroglu and Dihkan [23] used Landsat imagery and an automated image analysis technique to extract shorelines in salt marshes in Turkey. White and Madsen [24] used imagery collected by a balloon to map marsh landcover and ecological zones. With this study, we introduce two new methods that can be applied to high resolution (submeter GSD) datasets of elevation and imagery to identify marsh shorelines. Both methods have been designed to be much more effective and efficient than heads-up digitization from imagery, and to provide more objective and reproducible products.

Efficient and objective determination of the marsh shoreline bears similarity to the determination of open coast sandy shorelines. In the past, visual proxies like the high water line were used to map the shoreline [25,26]. However, these proxies were found to be difficult to discern, subjective and influenced by recent wave conditions [27]. The shoreline is now often defined as the mean high water (MHW) elevation and is often extracted from lidar (light detection and ranging) elevation data [28–30]. MHW elevations can be found in [31] or calculated with the VDatum tool (https://vdatum.noaa.gov/about.html). A MHW shoreline is often found by first creating a coast following baseline (i.e., reference line), then defining transects perpendicular to the baseline (although other methods exist that do not use baselines and transects, for example [20,32]). The elevation along these transects is found from lidar (or other elevation) data and then the location of the MHW elevation is found on each transect [33]. This leads to a datum-based, objective, and reproducible shoreline. Consistent estimates of shoreline change can then be calculated along these transects [34]. National scale efforts to compile shoreline position data have been successful for long, linear stretches of coast where historic maps and air photos can be heads-up digitized to complement the MHW elevation shorelines [35–37]. Back-barrier bays and marsh complexes often have highly crenulated shorelines with fine-scale features that are challenging and time consuming to digitize. These new marsh extraction methods provide a viable workflow to bridge gaps in national shoreline assessment efforts.

Unfortunately, a datum-based shoreline does not work as well in salt marsh environments since the salt marsh shoreline is the edge of the marsh vegetation, and the height of this varies [38]. Although the growth of many species of salt marsh vegetation are tidally controlled, the exact heights vary from one marsh to another due to differences in water levels that result from variations in local geomorphology [39,40]. We therefore developed the marsh edge from elevation data (MEED) method to calculate the "marsh scarp" which is the abrupt elevation change at the edge of most salt marshes [41–43]. The MEED method calculates slope from elevation data and defines the marsh scarp to be the maximum slope between mean high water and the mean tide level. The method can use elevation data from either lidar or structure from motion (SfM) aerial photogrammetry collected by unmanned aircraft systems (UAS, also known as unmanned aerial vehicle or drone).

We also developed a second method to locate the salt marsh shoreline. This method is called marsh edge by image processing (MEIP) and uses image classification techniques to find the edge of the salt marsh vegetation. Remote sensing and aerial imagery have been used to inventory and classify coastal wetlands for many decades [44,45]. Early examples of mapping wetlands in support of tidal wetlands legislation go back to the early 1970s when inventories were based on 1:12,000 scale color and/or color-infrared aerial photographs [46]. Within the same decade, the U.S. Fish and Wildlife Service established a National Wetlands Inventory (NWI) to provide information on the type and

distribution of wetlands nationwide [47]. Initially started at a scale of 1:250,000, NWI now produces 1:24,000 scale digital maps and also publishes status and trends reports every decade [48]. Depending on the application and the data availability, multispectral or hyperspectral imagery, lidar, and radar systems can be used for mapping coastal marshes [49]. For example, combining hyperspectral imagery with lidar-derived elevation has been shown to significantly improve the accuracy of mapping salt marsh vegetation [50]. The most commonly used methods in image classification are pixel-based and object-based classifications, although novel methods keep emerging [49]. Pixel-based-classification is based on the spectral information at individual pixels, whereas object-based classification also considers the relationship between contiguous pixels [51,52]. A combination of pixel-based classification with either supervised (where a training set is established prior to classification) and/or unsupervised classification methods is commonly found in the related literature [12,53–56]. In light of the literature and because of its simplicity to implement, we present the MEIP method, which combines multispectral imagery with elevation data and applies unsupervised pixel-based classification to define vegetated areas as an alternative approach for marsh edge detection. The MEIP method applies common classification techniques, utilizing higher resolution (1 meter) elevation and four-band imagery products that are consistent and available at a national scale. The same method can also be used to process even higher resolution data from smaller scale, rapid deployment systems such as UAS.

In this paper we show the results of the MEED method using lidar data from Massachusetts (MA), USA. We compare the results of both the MEED and MEIP methods using UAS data from the Plum Island Estuary in Massachusetts. The edge of the vegetation in the UAS data was heads-up digitized and compared to the MEED and MEIP results. The MEIP method is also used to calculate a shoreline for Plum Island Estuary using imagery from the National Agriculture Imagery Program (NAIP) and elevation data from the U.S. Geological Survey (USGS) Coastal National Elevation Database (CoNED). This paper ends with a discussion about the advantages and disadvantages of each method.

## 2. Materials and Methods

### 2.1. Study Area

We developed and tested the MEED method using lidar data from the coastal salt marshes of Massachusetts (MA), USA (Figure 1a). These marshes have a wide range of tidal environments, from micro to macrotidal, with a variety of morphologies ranging from fringing marshes to extensive channelized complexes. Both the MEED and MEIP methods were tested using data collected at the salt marsh network of the Plum Island Estuary, Massachusetts (Figures 1a and 2). This macrotidal marsh system is bordered by the Plum Island barrier beach to the east, the Merrimack River to the north, and Plum Island Sound to the south. It has extensive channelized marshes expanding west (landward) towards the Rowley and Ipswich Rivers, among other tributaries. Recent studies in the area [57,58] have documented the importance of lateral erosion and channel widening on the trajectory of the salt marsh system.

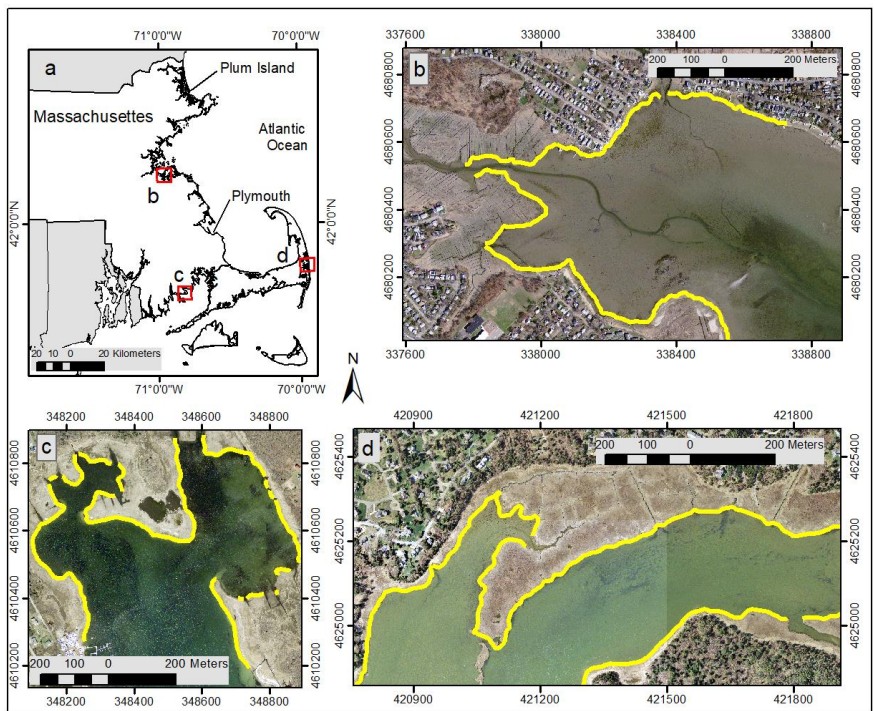

**Figure 1.** (**a**) Coastal Massachusetts, (**b**) Broad Meadows Marsh in Quincy, (**c**) Brant Island Cove in Buzzards Bay, (**d**) Pleasant Bay in Orleans. Yellow line in (b–d) shows marsh scarp from marsh edge from elevation data (MEED) using lidar data. Imagery from the U.S. Geological Survey (USGS).

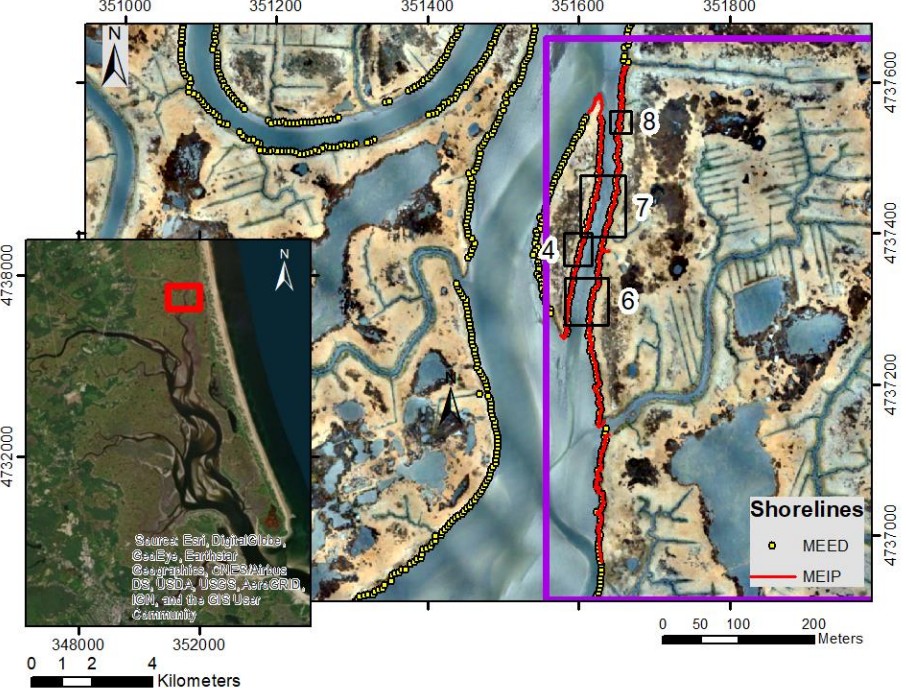

**Figure 2.** Plum Island Estuary, Massachusetts (MA). The shoreline calculated using the MEED method with unmanned aircraft system (UAS) data is shown with yellow dots. The unvegetated/vegetated line calculated using the marsh edge by image processing (MEIP) method with UAS data is shown as a red line. Box 4 indicates the area plotted in Figure 4. Box 6 indicates the area plotted in Figure 6. Box 7 indicates the area plotted in Figure 7. Box 8 indicates the area plotted in Figure 8. The purple line indicates the extent of the multispectral imagery used by the MEIP method. Imagery in main map is from UAS; imagery in the inset is from Esri.

## 2.2. Data

We developed the MEED method using a lidar survey collected by the U.S. Geological Survey (USGS) from 16 November 2013 to 27 December 2014, covering the entire coast of Massachusetts [59]. The survey was flown within 2 hours of low tide. We downloaded the point cloud data from NOAA (https://coast.noaa.gov/dataviewer/#/lidar/search/) in Universal Transverse Mercator (UTM), Zone 19, relative to the North American Datum of 1983 (NAD83) and the North American Vertical Datum of 1988 (NAVD88), using the geoid model GEOID12B. The vertical accuracy of this lidar survey was estimated to be 0.101 meters (95% confidence). To verify our results we used imagery collected by the USGS at the same time as the lidar data [60].

In addition to the lidar data, we also used data collected by a UAS flown over about 1.5 km$^2$ of the salt marsh network of the Plum Island Estuary, Massachusetts (Figure 2) on 14 November 2017 [61]. Low altitude (80 and 100 meters above ground level) digital images were taken using a Ricoh GRII digital camera on a 3DR Solo UAS. The marsh was surveyed within 2 hours of low tide and around solar noon (to reduce shadows). These data were processed using structure from motion (SfM) to produce high quality, high resolution elevation data. We used Agisoft PhotoScan Professional (v1.2.6, https://www.agisoft.com/) and followed the standard workflow [62–65] to produce digital elevation models (DEM) that are spatially dense, approximately 5 centimeters and have an uncertainty of approximately 10 centimeters. These data are also in UTM, Zone 19, NAD83, NAVD88.

The MEED method for calculating the location of the marsh scarp requires knowledge of mean high water (MHW) and mean tide level (MTL). Whenever possible, previously published data [31] was used to obtain MHW. In areas not covered by this work, VDatum (https://vdatum.noaa.gov/about.html) was used to find MHW. Following the method laid out by Weber et al. [31], one representative value of MHW was used for each marsh system, even though VDatum provides a continuously varying estimate. VDatum was also used to obtain MTL. We used an average value of 0 meters for MTL for all of Massachusetts.

For part of the study area in Plum Island Estuary, the UAS flew with a MicaSense RedEdge multi-spectral camera that captures five specific bands of the visible spectrum (blue, green, red, red edge, and near-infrared) (Figure 2). For this study, an area of about 0.5 km$^2$ was flown with the multispectral camera. These data were used to test the MEIP method. The MEIP was also tested using National Agriculture Imagery Program (NAIP) and USGS Coastal National Elevation Database (CoNED) datasets from Plum Island. The NAIP dataset consists of 4-band (blue, green, red, infrared), 1 meter resolution imagery (http://www.fsa.usda.gov/programs-and-services/aerial-photography/imagery-programs/naip-imagery). This imagery was already rectified and corrected for alignment. For this study area the CoNED dataset also had a 1 meter resolution (https://topotools.cr.usgs.gov/coned/).

## 2.3. MEED Methodology

The MEED method begins with a DEM (Figure 3a). When lidar data were used, the LAS data files (the standard lidar file format) were loaded into ArcMap and a 1 meter resolution DEM was constructed using an average binning method (averaging elevation of all the data points within the cell as the elevation for each cell). When data from the UAS were used, the DEM created by the SfM processing was used. These DEMs were more dense, approximately 5 centimeters. Previous studies have discussed the difficulty of getting a true bare Earth DEM in a salt marsh because the laser inconsistently penetrates the marsh grass [21,66,67]. This is less of a concern with the MEED method because the edge of the marsh platform is identified with maximum change in slope.

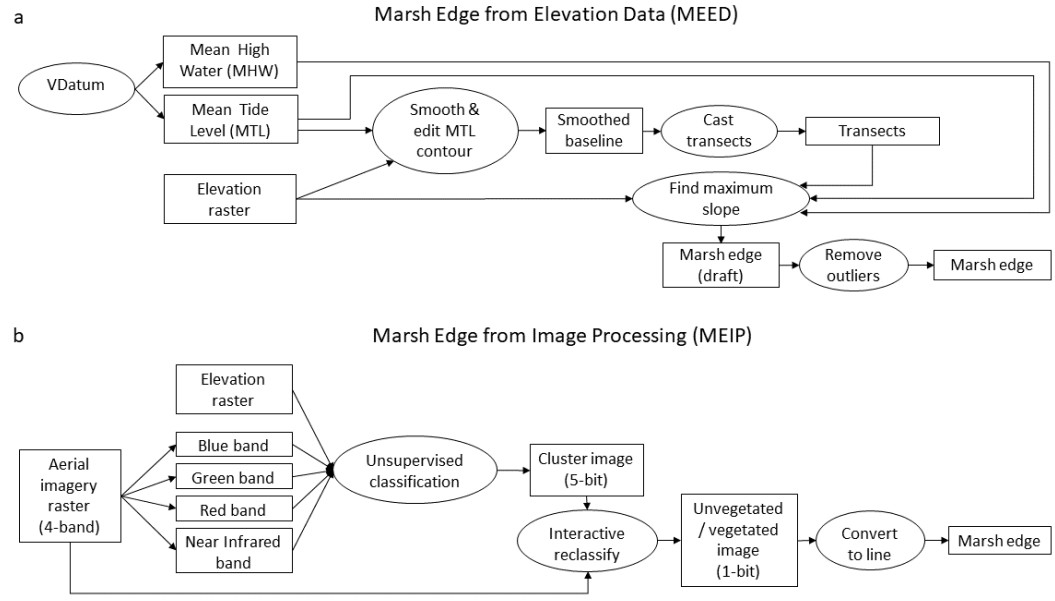

**Figure 3.** Flow charts for (**a**) marsh edge from elevation data (MEED) and (**b**) marsh edge from image processing (MEIP) methods.

The DEM was used to contour the MTL. The resulting contour was smoothed with the polynomial approximation with exponential kernel method (PAEK) [68] and a smoothing tolerance parameter of 50 meters. Occasionally the results were improved by contouring a slightly different elevation or by using a slightly different smoothing length (depending on the crenulation of the marsh edge). The goal was to get a relatively smooth line that generally followed the edge of the marsh; since this line was only to be used as a baseline, its exact location and shape were not critical. Using the ArcMap editing tool, sections of the line were removed from areas that were not actually part of the marsh shoreline (for example around small pools on the marsh platform) or areas that were too narrow for this method to extract a shoreline (for example along drainage ditches <10 meters wide). Where appropriate, disconnected sections of the line were joined with the editor. The edited line was converted to points and exported as a comma separated values (CSV) file. The DEM was also exported as an ascii file so that it can be easily loaded into Matlab©.

Both the DEM and the smoothed MTL contour were loaded into Matlab©. The MTL contour was used as a baseline, and equally spaced (5 meter intervals) points were defined along it. At each point, transects perpendicular to the baseline were created using Matlab scripts written by one of the authors of this paper (Figure 4a). The transects extended in both directions from the baseline and were usually about 30 meters long. Transects were lengthened if the baseline was far from the edge of the marsh or shortened if the transects intersected with unwanted parts of the marsh. The elevation along each transect was extracted from the DEM and the slope was calculated between all the points along each transect. The scarp was identified as the point of maximum (in magnitude) slope between MHW and 0.5 meters below MTL (Figure 4b). An offset of 0.5 meters below MTL was used because sometimes valid solutions were below MTL. Because the code occasionally placed the scarp at steep offshore or onshore features, the scarp was displayed over imagery and the outliers were manually removed.

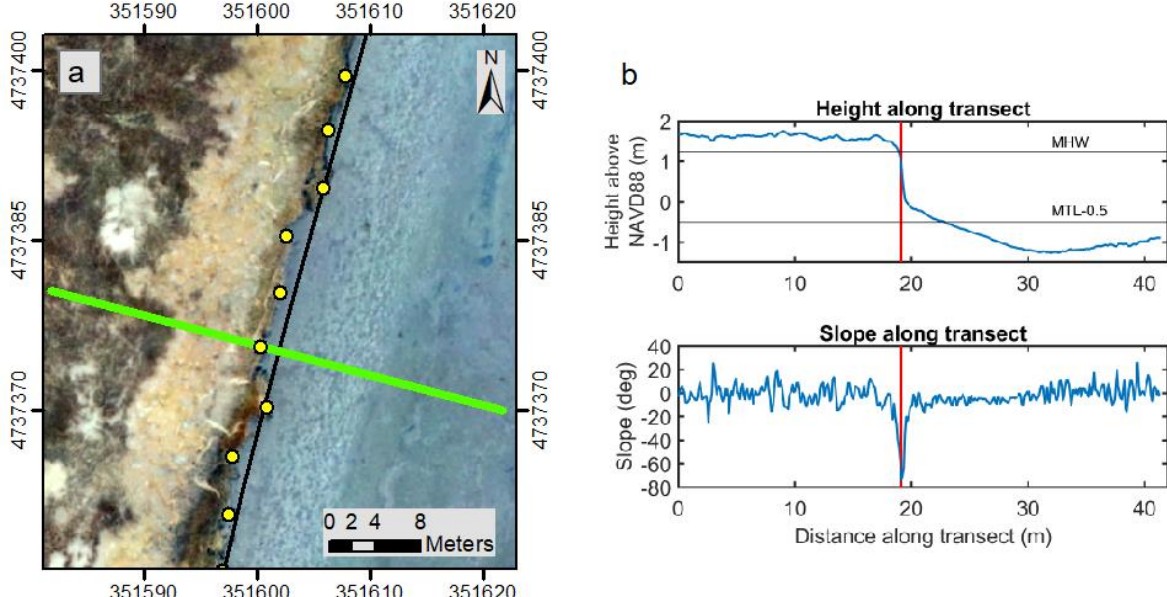

**Figure 4.** (**a**) UAS imagery from Plum Island, MA. The black line is the smoothed mean tide level (MTL) contour that is being used as a baseline for the marsh edge from elevation data (MEED) method, the green line is a sample transect, the yellow dots are the calculated marsh scarp; (**b**) UAS elevation data from the sample transect shown in Figure 4a. The vertical red line is the location of the calculated scarp. Top: Elevation along transect, the horizontal black lines indicate mean high water (MHW) and (MTL - 0.5 meters). Bottom: Slope along transect.

## 2.4. MEIP Methodology

In the MEIP method, elevation and imagery datasets were used together to calculate an unvegetated/vegetated line (Figure 3b). An unsupervised classification tool (Iso cluster unsupervised classification in ArcMap) that combines iterative self organizing data analysis technique (ISODATA) clustering [69] and maximum likelihood classification [70] tools was applied to a set of raster datasets that consisted of the elevation dataset and the four bands (blue, green, red and near infrared) from the imagery being used (in this study, either UAS or NAIP). Before feeding into the classification algorithm, elevation values were rescaled to cover the same range of possible values (0–255) with the 8-bit imagery. The algorithm started with assigning arbitrary cluster centers in the multidimensional parameter space and iteratively migrated them to minimize the mean Euclidian distance to each cell. The classification algorithm was used to produce an image with a maximum of 32 classes. However, in some cases a smaller number of classes were obtained. For example, if any classes were statistically similar after the classification became stable, the algorithm automatically merged them into a single class. Also, if a class had fewer cells than a threshold value, it was automatically eliminated at the end of iterations. The threshold value, set to 5000 cells in this case, was usually multiple orders of magnitude smaller than the size of the underlying image. After the unsupervised classification, the resulting raster was visually compared to the 3-band image (red, green, blue (RGB)) and the ArcGIS World Imagery Basemap (acquired June 2017) to further classify it into vegetated and unvegetated classes. The raster was then converted to polygons of unvegetated and vegetated regions. Polygons created with this method mark marsh shoreline and unvegetated/vegetated lines inland. For this study, only the shoreline part was used after converting the polygons to lines.

## 2.5. Heads-Up Digitized Shoreline

The UAS imagery was loaded into ArcMap and the edge of the vegetation was traced manually. We tried to be as precise and accurate as possible, but it was often difficult to determine the exact edge

of the vegetation. Some parts of the imagery were not very clear and the vegetation is often a similar color as the substrate.

## 3. Results

Using the MEED method with lidar data, we calculated a marsh scarp for many of the coastal salt marshes in Massachusetts; these shorelines are available for download at the USGS ScienceBase Catalog [71]. A visual inspection of the line plotted over the imagery collected at the same time as the lidar data [60] revealed that the line generally tracks the edge of the vegetation (Figure 1b–d). Cases where the marsh scarp did not fall at a vegetation edge were extremely limited. In most of these cases it was difficult to determine the "true" marsh edge from the imagery (Figure 5).

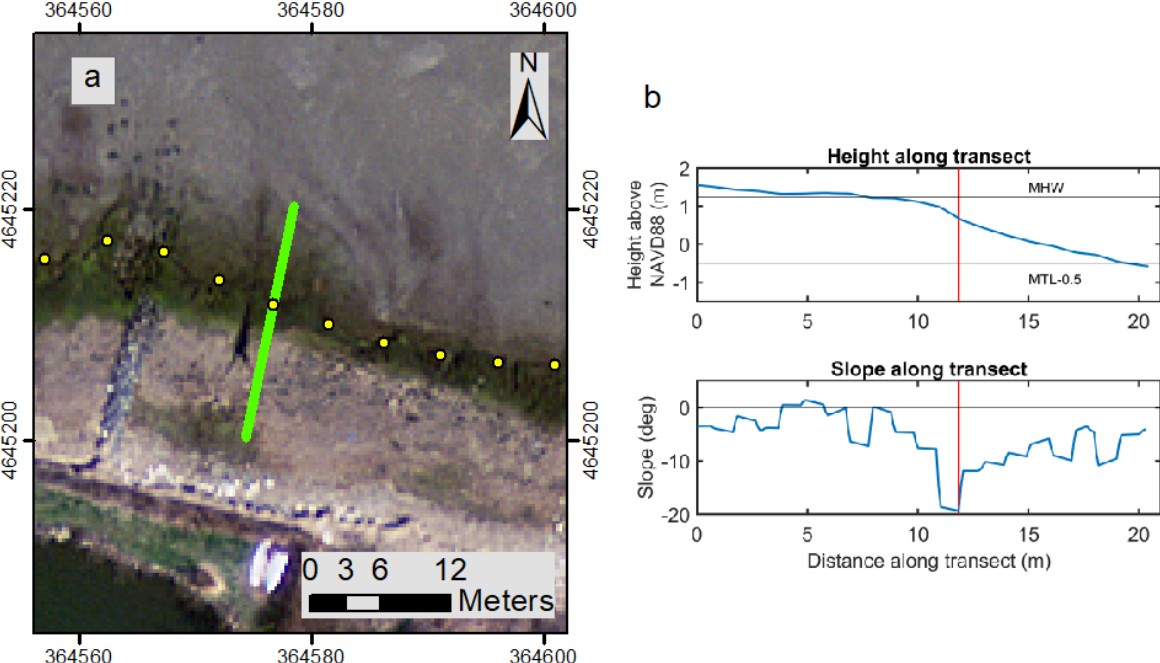

**Figure 5.** (**a**) Plymouth, MA, USGS imagery. Yellow dots: Marsh scarp from MEED, green line: Sample transect; (**b**) lidar elevation data from transect shown in Figure 5a. The vertical red line is the calculated marsh scarp. Top: Elevation along transect, the horizontal black lines indicate MHW and (MTL - 0.5 meters). Bottom: Slope along transect.

The MEED method was used to calculate the marsh scarp using UAS data from the Plum Island Estuary in MA (Figure 2). The marsh scarp usually closely follows the edge of the vegetation seen in the imagery (Figures 4a, 6a and 7a). Multispectral imagery was collected for a smaller area of Plum Island and from this, the unvegetated/vegetated line was calculated using the MEIP method. This line also closely follows the edge of the vegetation seen in the imagery (Figures 6a and 7a). For the 850 meters of marsh shoreline where we have results from both methods, the root-mean-square deviation between the marsh scarp and the unvegetated/vegetated line was 0.6 meters. For the transect shown in Figure 4, the agreement between the two methods is within 2 centimeters. Very occasionally the marsh scarp does not fall at the edge of the vegetation and the two methods may give different results (Figures 8 and 9). Two local maxima (in magnitude) are visible in the slope (Figure 8b). This occurs rarely and when it does, usually the larger (in magnitude) of the two slope minima corresponds with the edge of the vegetation, but on rare occasions, as can be seen here, it does not. In this case, the horizontal offset between the two local minima is 1.1 meters.

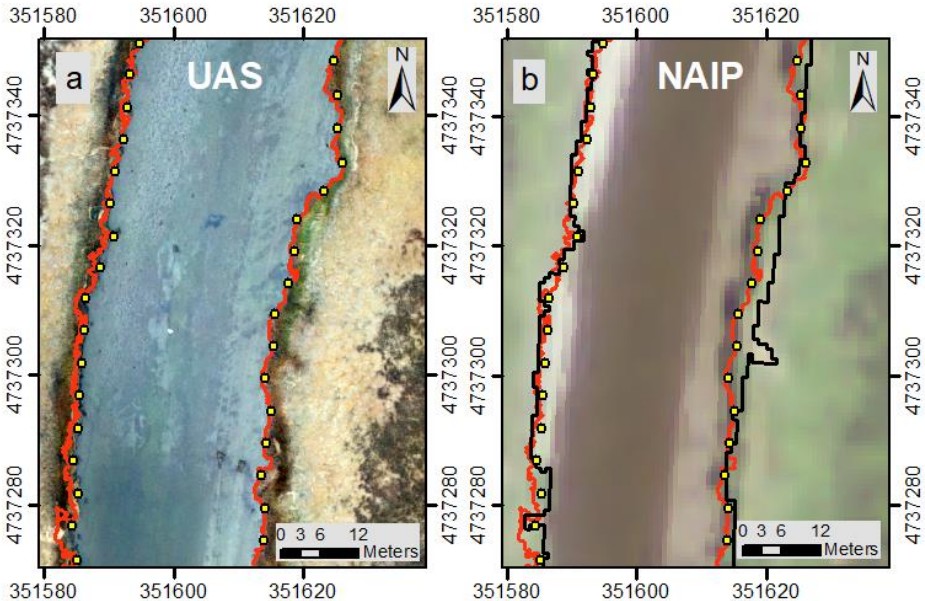

**Figure 6.** Plum Island, MA (Box 6 in Figure 2) showing (**a**) imagery from UAS; (**b**) imagery from the National Agriculture Imagery Program (NAIP). Yellow dots: Marsh scarp from MEED, red line: Unvegetated/vegetated line from MEIP using UAS data, black line: Unvegetated/vegetated line from MEIP using NAIP imagery.

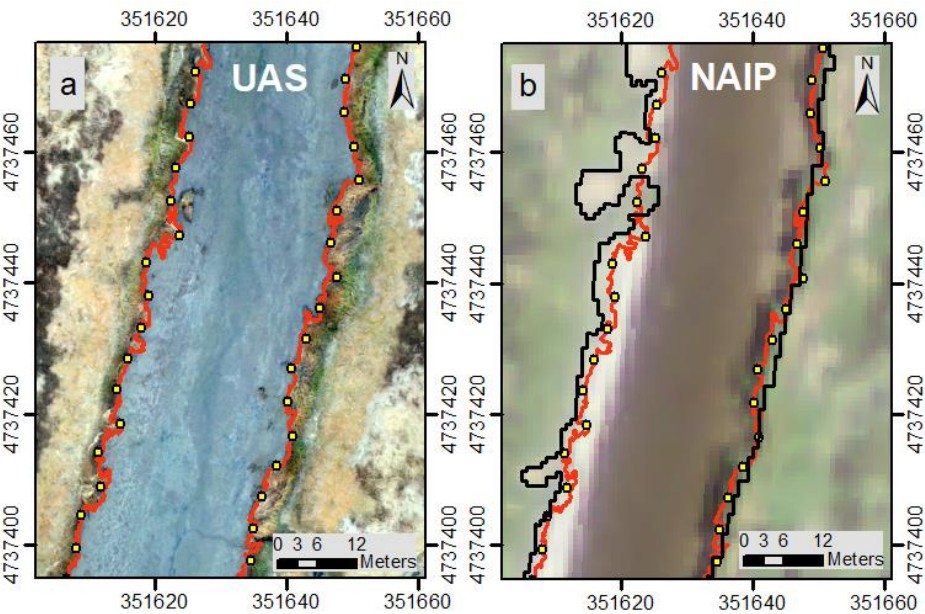

**Figure 7.** Plum Island, MA (Box 7 in Figure 2) showing (**a**) imagery from UAS; (**b**) imagery from NAIP. Yellow dots: Marsh scarp from MEED, red line: Unvegetated/vegetated line from MEIP using UAS data, black line: Unvegetated/vegetated line from MEIP using NAIP imagery.

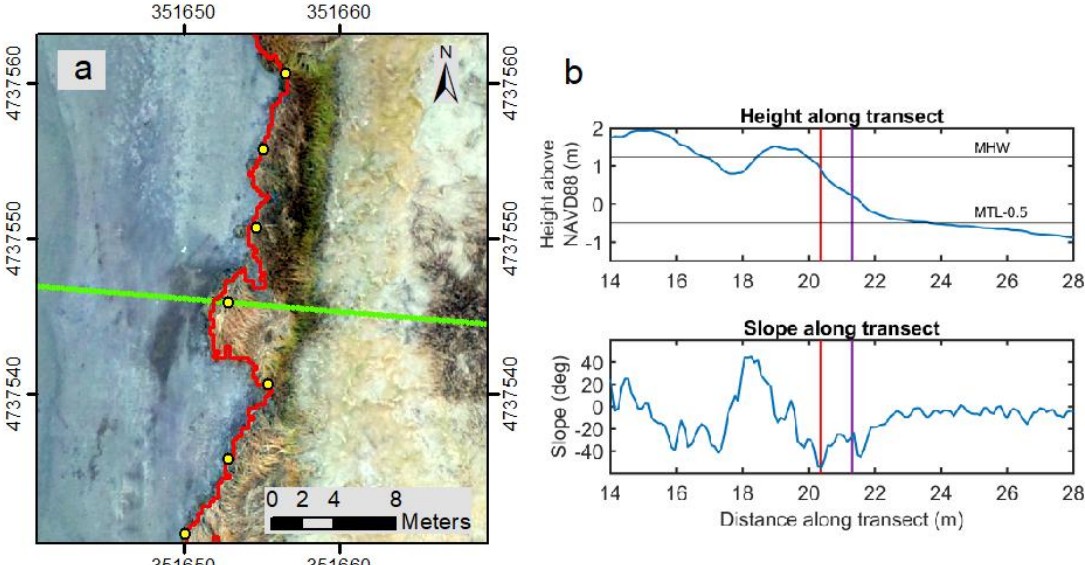

**Figure 8.** An example of a transect where the scarp and unvegetated/vegetated line disagree in Plum Island, MA. (**a**) UAS imagery, yellow dots: Marsh scarp from MEED, red line: Unvegetated/vegetated line from MEIP, green line: Sample transect shown in Figure 8b; (**b**) UAS elevation data from the sample transect shown in Figure 8a. Top: Elevation along transect, the horizontal black lines indicate MHW and (MTL - 0.5 meters). Bottom: Slope along transect. Red line: The marsh scarp from MEED, purple line: Unvegetated/vegetated line from MEIP. The scarp and unvegetated/vegetated line are 0.97 meters apart; the two local maxima (in magnitude) of slope (between MHW and MTL - 0.5 meters) are 1.1 meters apart.

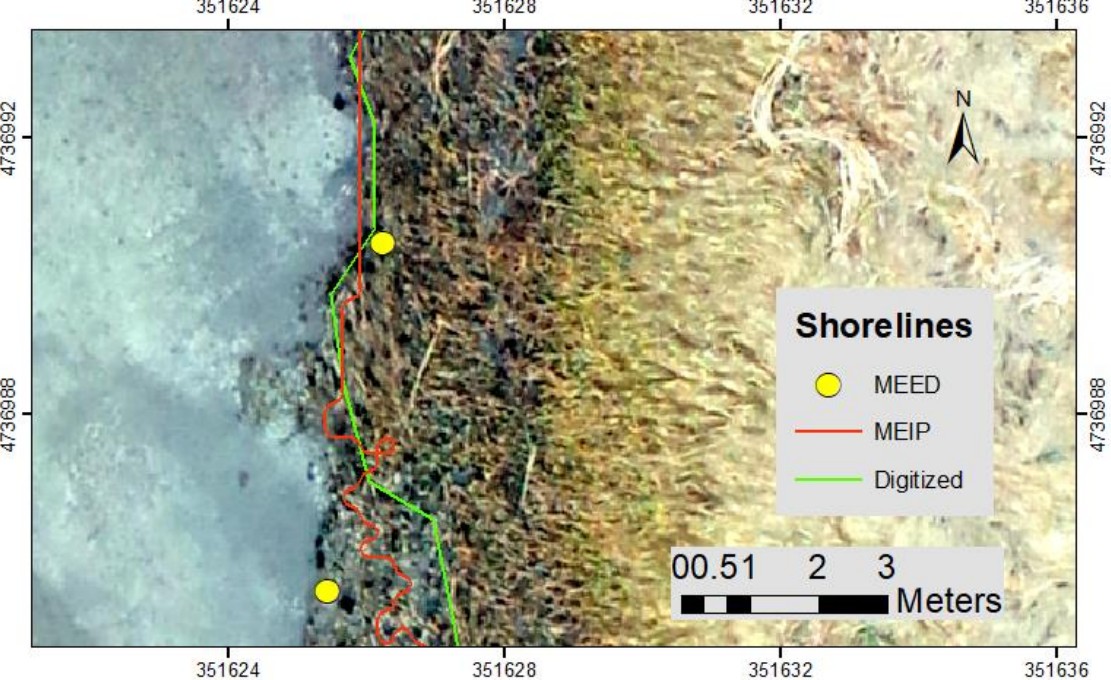

**Figure 9.** Plum Island Estuary, MA, UAS imagery. Yellow dots: Marsh scarp from MEED, red line: Unvegetated/vegetated line from MEIP, green line: Digitized edge of vegetation. The challenge in determining the location of shoreline with the heads-up digitization can be seen in the lower half of the image where the operator selected a somewhat subjective marsh shoreline, MEED method identified the maximum slope, and MEIP method identified a location somewhere between the two.

The heads-up digitized edge of vegetation was compared to the MEED and MEIP results. The root-mean-square deviation between the MEED and the digitized shoreline was 0.43 meters. The root-mean-square deviation between the MEIP and the digitized shoreline was 0.40 meters. These differences reflect the fact that sometimes additional error was introduced with the heads-up digitalization because it was not always possible to identify the edge of vegetation precisely in the visual inspection of imagery (for example in the case of slumped marsh edge or marsh die-back, Figure 9) which led to some additional uncertainty.

The NAIP data and CoNED datasets were also used as inputs to the MEIP method to calculate the unvegetated/vegetated line. The resulting line was compared to both the scarp calculated from the MEED method using UAS data and the unvegetated/vegetated line calculated from the MEIP method using UAS data. The line found using NAIP data did not follow the others as closely (Figures 6b and 7b). This is partially because the resolution of the NAIP and CoNED data is 1 meter, in contrast to the UAS resolution of 5 centimeters. Another factor might be that the NAIP imagery was flown on 12 July 2017 when aboveground live biomass was present (i.e., green), whereas the UAS data was flown 14 November 2017 when the marsh grass had senesced (i.e., brown). For the 850 meters for which we have data from both data sets, the root-mean-square deviation between the unvegetated/vegetated lines extracted from the UAS and NAIP data was 1.74 meters. The line that was calculated from the NAIP data was slightly onshore of the line calculated from the UAS data, with a mean difference between them of 0.58 meters.

## 4. Discussion

It is advantageous to have multiple methods with which to calculate the marsh shoreline. Data availability can limit what method can be used. The type of shoreline required, and the end-user application can also determine which method would be the best. Calculating the marsh shoreline using more than one method can be a way to estimate uncertainty, as we do in this paper. This is useful as ground-truthing can be difficult.

The marsh scarp is a useful proxy for a salt marsh shoreline because salt marsh vegetation usually ends at a distinct drop in elevation. Therefore, the edge calculated by the MEED method usually closely follows the edge of vegetation seen in imagery (Figures 4a, 6a, 7a, and 8a). The marsh scarp has been noted in other studies, for example in Figure 5 of [72] the marsh scarp ("low marsh ramp" in their study) is evident in a generalized profile from the Plum Island Estuary. This feature is also highlighted in Figure 4 of [57], which shows cross-shore profiles in areas that have undergone erosion. Goodwin et al. [73] used scarps to differentiate marsh platforms from tidal flats. They developed a new method to identify scarps in a DEM that they call topographic identification of platforms (TIP).

The abrupt elevation change between a salt marsh plain and an intertidal flat results from the stabilizing influence of marsh vegetation, and subsequent vertical accretion due to inorganic sediment trapping and autochthonous organic material deposition. Unvegetated flats are subjected to relatively higher bed shear stresses due to waves and currents, while the vegetated plain is protected due to increased drag and attenuation of waves. This leads to a natural evolution of an elevated marsh plain and lower unvegetated, intertidal flat [74,75], which can be successfully detected with the MEED method. While there are instances of gently sloping marsh estuary transitions, these would typically be found in low energy environments where salt marshes are expanding on to uncolonized intertidal flats. Given persistent sediment deficits and sea level rise [11,12], these more challenging situations for the MEED method are likely to be less common. A few gently sloping marsh estuary transitions were seen in this dataset. In these cases, there was no scarp and no distinct edge of vegetation (Figure 5). In these situations, the MEED method still provides a shoreline that can be obtained repeatedly with a consistent method.

The MEED method worked well with this data collected in November when the marsh vegetation was wilted and mostly flat. It should also work in the spring and summer when the marsh grass is growing and thus the marsh scarp seen in the DEM is even larger. The MEED method has many

advantages, including allowing for the calculation of other potentially useful quantities, such as scarp slope or scarp width. The MEED method can also be modified to identify both the edge of the marsh platform and the seaward edge of any cleaving blocks. Another useful quantity that can be calculated using the MEED method is the height of the marsh edge. This can be important because some studies investigate the relationship between elevation and types of marsh vegetation.

There are some drawbacks to the MEED method. It is not completely automated, the smoothed MTL must be manually edited to get a useful baseline and the final shoreline should be visually checked for outliers. These outliers occur when the transects cross steep features far from the marsh scarp. Another drawback is that the scarp can be several meters wide and there may be more than one local maximum in slope along the scarp (Figure 8). Picking the maximum slope is objective and repeatable, but perhaps the location of the maximum slope may move over time, even if overall the scarp itself does not shift landward. Finally, because the MEED method uses a baseline, it is poorly suited to calculate the shoreline along narrow (~<10 meters) creeks. However, the use of a baseline makes future calculations of shoreline change straightforward.

Because the MEIP uses the infrared band, it also worked well with this data collected in November when the senesced marsh vegetation was sometimes hard to differentiate from sediment in the visual imagery. The MEIP method is mostly automated and in theory it can be applied to large domains until limited by the computational power. Practically, however, as the image size gets larger the interactive part of the method (e.g., final reclassification step) is likely to become more time consuming as there will be more areas over which to verify the final classification. The advantage of the interactive part, on the other hand, is that it gives more control to the operator, thus making the method very flexible. This hybrid approach between a fully automated and an interactive method is less subjective than a fully interactive method where shorelines may be digitized manually. In most cases it is also more practical than a fully automated method because the fully automated methods usually require a large training set and/or many tests to attain proven confidence or otherwise they still require an evaluation of the final product by the operator. In contrast to methods that require a priori training dataset, in MEIP, the decision making happens at the final step. This has two main advantages. Firstly, the evaluation of the classification happens concurrently with the classification because of the interactive nature of this final step. Secondly, in the case of a misclassification, the operator has the option to prioritize correct classification of areas of interest over others (e.g., in the case of marsh shoreline detection, if reclassifying a subclass as water results in a more accurate shoreline dataset, the operator may choose to disregard misclassification of some of the interior features such as salt panes or sand deposits instead of starting from scratch).

The MEIP method can also be used for other edge detection purposes at the interactive stage (e.g., edge detection for vegetation other than marsh, water edge detection etc.). Because of many factors such as turbidity in water, mud flats and wilting marsh vegetation it is sometimes not possible to differentiate water in 3-band imagery. The infrared band available in the 4-band imagery used by MEIP is absorbed by water thus enhances detection of water. Including elevation data, in addition to imagery, introduces another layer of information, which tends to increase the detection of very narrow creeks. This is usually true when the width of a creek is close to the raster cell size, therefore, the improvement is a function of the size and extent of creeks and the resolution of the elevation dataset. A limitation of the MEIP method is that it performs better with the 4-band imagery, which may not always be available.

## 5. Conclusions

Given the ubiquity of lidar data and the need for a quantitative measure of salt marsh shorelines, we developed a method to calculate the marsh shoreline from remotely sensed elevation data. This method, called marsh edge from elevation data (MEED), calculates the location of the marsh scarp as a proxy for the marsh shoreline. The marsh scarp is the abrupt elevation change that usually characterizes the edge of a salt marsh. The MEED method can use either lidar data or data from an

unmanned aircraft system (UAS, drone) using structure from motion processing. The MEED method was developed using lidar data from Massachusetts, USA, and UAS data from the Plum Island Estuary in Massachusetts. The method is accurate; the marsh scarp closely follows the edge of vegetation seen in imagery. Because the MEED method is mostly automated, it can be efficiently applied to define marsh shorelines for large domains. Because the method uses baselines and transects, calculation of shoreline change is straightforward.

Because high resolution multispectral imagery is becoming more widely available (e.g., from National Agriculture Imagery Program (NAIP) available at 1 meter resolution, nationwide; or from UAS measurements at higher resolutions, but with localized coverage and geospatially scattered projects), we also developed a method to use multispectral imagery together with elevation data (e.g., from the Coastal National Elevation Database (CoNED) at 1 meter resolution, nationwide or from UAS measurements at higher resolutions) to calculate the edge of the vegetation. This method, called marsh edge by image processing (MEIP), produces an unvegetated/vegetated line that can be used to locate the marsh shoreline. Because it is based on an image classification algorithm, it can also be used in applications other than shoreline detection. The MEIP method is a combination of a fully automated classification and interactive reclassification methods and is therefore efficient when applied to large domains, but also flexible to support minor adjustments at the final stage of interactive reclassification. This method was also tested using UAS data and NAIP data from Plum Island Estuary, Massachusetts and fulfilled expectations after visual inspection against the underlying imagery.

We quantitatively compared the marsh scarp using the MEED method with the unvegetated/vegetated line from the MEIP method using UAS data from the Plum Island Estuary in Massachusetts. For the 850 meters of shoreline, for which we have results from both methods, there was a root-mean-square deviation of 0.6 meters between the two methods. We also compared both lines to a heads-up digitized line, there was a root-mean-square deviation of 0.43 and 0.40 meters between the digitized line and the MEED and MEIP methods, respectively. We also tested the effect of source data on the results of the MEIP method by using NAIP data. The root-mean-square deviation between the unvegetated/vegetated line calculated from the NAIP data and the UAS data was 1.74 meters. Both methods proved to be effective and efficient in identifying salt marsh shorelines within the study area, and they should be tested in other regions to assess their applicability in different geographic settings.

**Author Contributions:** A.S.F. developed the MEED method. Z.D. developed the MEIP method. Both A.S.F. and Z.D. wrote the manuscript. N.K.G. assisted with the development of both methods and the writing of the manuscript.

**Funding:** This project was supported by the U.S. Geological Survey (USGS) Coastal/Marine Natural Hazards and Resources Program as well as the Massachusetts Office of Coastal Zone Management under interagency agreement 16ENMALQ006000.

**Acknowledgments:** We thank the U.S. Geological Survey Woods Hole Aerial Imaging and Mapping Group for all their excellent work: Sandra Brosnahan, Elizabeth Pendleton, Seth Ackerman and Emily Sturdivant. We also thank Julia Knisel and Dan Sampson at the Massachusetts Office of Coastal Zone Management for their support. We are grateful to Kathy Weber and Christie Hegermiller for their helpful reviews of the manuscript. Three anonymous reviewers made many excellent suggestions that substantially improved the manuscript.

**Conflicts of Interest:** The authors declare no conflict of interest. The funding sponsors had no role in the design of the study; in the collection, analyses, or interpretation of data; in the writing of the manuscript, and in the decision to publish the results. Any use of trade, firm, or product names is for descriptive purposes only and does not imply endorsement by the U.S. Government.

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
