# Peer review of "Identifying Salt Marsh Shorelines from Remotely Sensed Elevation Data and Imagery"

_remotesensing, doi:10.3390/rs11151795_

Round 1

Reviewer 1 Report

SUMMARY and COMMENTS:

This article describes two methods developed using readily available data to more efficiently delineate shorelines in salt marshes. These methods combine previously established techniques to reduce the subjectivity and time intensive nature of earlier methods. The article is fairly well organized, however I did find having to refer back and forth between multiple figures a bit cumbersome. Not all methodology is clear. This could easily be addressed by adding some clarifying information - especially related to data specifications, how data are generated and passed from one process to another. I also found that the article lacked some basic introductory information related to the importance of wetlands, the needs for assessing long- and short-term shoreline change, and  existing methods/tools (and their limitations) for delineating wetland shorelines. Although the article focuses on readily available data and identifies some limitations of each method, it would be useful if the authors provide additional ways to overcome those limitations, and/or provide future research/advancements (using multispectral/hyperspectral imagery and vegetative indices with water indices, or object based image analyses which could increase classification efficiency and accuracies).

GENERAL:

Title - Is "Identifying Salt Malt Marsh..." correct? Or should it be "Identifying Salt Marsh Shorelines..."

Suggestion - It could be useful to have more information about DSAS and how these data could better assist in evaluating historical and more recent shoreline changes.

Inconsistencies - sometimes listed as "i.e." sometimes as "i.e.,"

Acronyms - always spell out first occurance

LINE15 - Should it be "marsh scarp." and not "marsh scarp".?

LINE24 - spell out "meter"

LINE45 - Spell out first occurance of LiDAR

LINE46 - Should VDaatum be "VDatum"?

Figure 1 - Is "Massachussettes" spelled incorrectly in Panel A?

Figure 1 caption - semicolon after (a), but commas used after b, c, and d. Is that correct?

LINE120 - What were the water levels at the time of image/data acquisition?

LINE127 - UAS airframe and sensor specfications should be provided.

LINE138 - Check with journal requirements/standards, but it would certainly help the reader if the Authors are listed here ...."laid out by Weber et al. [19],"

LINE144 - "These data were used"...

LINE153 - delete "by" "DEM was constructed using an average"...

FIGURE 3 - Change Vdatum to "VDatum" for consistency

FIGURE 3 - Increase font size to make more legible

LINE160 - "Once the DEM was in ArcMap, the MTL was contoured" is an awkwardly worded sentence. 

LINE167 - "The DEM was also exported" is unclear. Consider the where and why questions here. 

LINE168 - Does MATLAB need to be cited?

LINE169 - It is uncertain how transects were created. Was it done in MATLAB? There are a number of GIS tools that automatically create transects of varying distance and length. Either state the process used or possibly point the reader to existing tools.

LINE 178 - I originally wondered about how these developed methods would work with mud flats, and submerged and floating aquatics. I later read how some of these were addressed in the conclusions, but it might be useful to add some discussion here.  li

LINE191 - Which imagery is this refering to? UAS or NAIP?

LINE199 - Was this a heads-up classification process? If so, was that then used to create training ROIs for the maximum likelihood classification? Some detail is needed for more clarity.

LINE206 - double occruance "of of" the coastal...

LINE209 - Marsh scarp not falling at vegetation edge were limited in Massachusetts, but might be more common elsewhere. Also, the authors briefly describe senescence, but how might seasonality or differences in plant senescence impact these methods?

LINE260 - Were the NAIP imagery checked for correct alignment? Were they rectified?

LINE275 - Same as above - check Journal requirements, but it seems awkward to start sentence with [55]. Suggest "Goodwin et al. [55]"

LINE275 - Is "TIP" an acronym? If so, spell out here.

LINE305 - Replace "in" with "it" ..."theory it can be" 

LINE314 - Should "a priori" be italicized?

LINE345 - Remove "NAIP"

LINE357 - Add period "." to end of sentence.

LINE421 - Stockdon, H.; Sallenger, a.; List - capitalize "A"

LINE440 - List all authors

Reviewer 2 Report

This is an interesting study presenting two methods of mapping salt marsh shoreline from remotely-sensed data. The paper is well written, the methods are clearly described, and I think there is a contribution to be made through this work. However, the paper does not adequately represent the current state of knowledge in shoreline mapping of salt marshes using remotely-sensed data. Several important references are missing, and related techniques and challenges discussed in other papers are not addressed. Below are the specific comments from my review:

1.      I do not fully agree with the statement on lines 35-36 that “there is no standardized method for delineating salt marsh shorelines using remotely sensed data.” NOAA NGS’s Coastal Mapping Program (https://www.ngs.noaa.gov/RSD/cmp.shtml) has standardized procedures for mapping the National Shoreline depicted on NOAA nautical charts and the NOAA Continually Updated Shoreline Product (CUSP) and available via Shoreline Data Explorer (https://www.ngs.noaa.gov/CUSP/). Specifically, the NGS CMP has well-developed procedures for mapping salt marsh shoreline, as well as outer coast sandy shoreline, vegetated shoreline, built shoreline, and other shoreline types using various types of remotely-sensed data, including tide-coordinated NIR aerial photography, airborne lidar, and multispectral imagery. Because the methods were developed for operational use, rather than research purposes, they are less well documented in research literature than they perhaps should be. However, general information on the procedures can be found in Graham et al. (2003), Leigh et al. (2005), White (2007), and White et al. (2011). This is not to say that there is no need for ongoing research into methods of salt marsh shoreline mapping from remotely-sensed data (such as this study), but I think the sentence in question clearly understates the current status of work in this area.

2.      A well-documented challenge of using lidar in salt marshes is the nearly ubiquitous elevation bias, and several studies have focused on overcoming this limitation (e.g., Hladik and Alber, 2012; Schmid, Hadley, and Wijekoon, 2011; Rogers et al., 2016). Was elevation bias observed in the lidar data collected over the salt marshes in this study? If so, how was it handled? And, if not, what acquisition or processing procedures were used to remove or reduce the bias?  

3.      The specifications and acquisition parameter settings for the remotely-sensed data should be listed. For example, what lidar system was used in collecting the data? What were the nominal (or measured) point density, acquisition altitude, laser wavelength, and pulse width (all important parameters in lidar shoreline mapping)? What geoid model was used in converting to NAVD 88 orthometric heights? What UAS (make/model) was used? What was the UAS flying height? What type of camera was installed on the UAS (make, model, focal length, spectral bands)? Did the UAS have “survey-grade” (i.e., carrier-phased based relative positioning: RTK or PPK) GNSS, or did it use code-based DGNSS, or standalone GNSS? How many photo control points were used in processing in Photoscan, and how were they surveyed?

4.      Lines 44-51: Another method of extracting MHW shoreline from airborne lidar data using VDatum and contouring algorithms is described in White, 2007 and White et al., 2011. A key characteristic of this method—and one that distinguishes it from the papers listed in this paragraph—is that it is not based on transects, so challenges related to transect spacing and selection of a baseline are eliminated (although, of course, there are pros and cons to all methods).

5.      A recommendation--perhaps for future work--would be to develop a total propagated uncertainty (TPU) model for your two shoreline mapping techniques using analytical uncertainty propagation (e.g., GLOPOV) and/or Monte Carlo methods. A side benefit would be a sensitivity analysis, enabling the most important component uncertainties to be identified.

6.      Although the paper is well written, I do have a few wording and formatting edits from my review:

a.       On line 61, you define UAS as “Unmanned Aerial Systems” and on line 338 as “Unmanned Autonomous System.” Per the FAA’s usage, UAS is “Unmanned Aircraft System.”

b.      The sentence on lines 314-321 is extremely long and difficult to follow. I would split it into at least two separate sentences.

c.       Line 288: there’s an unnecessary period before “(Figure 5)”

d.      Line 291: I would replace “like” with “such as”

References:

Graham, D., M. Sault, and J. Bailey, 2003. National Ocean Service shoreline: past, present, and future. Shoreline Mapping and Change Analysis: Technical Considerations and Management Implications, Journal of Coastal Research, Special Issue No. 38, pp. 14–32.

Hladik, C., and M. Alber, 2012. Accuracy assessment and correction of a LIDAR-derived salt marsh digital elevation model. Remote Sensing of Environment, Vol. 121, pp. 224-235.

Leigh, G.E. and J. Hale, 2005. Scope of Work Shoreline Mapping. National Oceanic and Atmospheric Administration. Online: https://www.ngs.noaa.gov/ContractingOpportunities/SOW_V13A.pdf

Schmid, K.A., B.C. Hadley, and N. Wijekoon, , 2011. Vertical accuracy and use of topographic LIDAR data in coastal marshes. Journal of Coastal Research, 27(6A), pp. 116–132.

White, S., 2007. Utilization of LIDAR and NOAA's vertical datum transformation tool (VDatum) for shoreline delineation. In Proceedings of IEEE OCEANS 2007, pp. 1-6.

Reviewer 3 Report

The study presents a method for identifying on important indicator of salt marsh extent and change. The study demonstrates multiple methods for utilizing elevation data or a combination of elevation and imagery to determine the marsh scarp. The study has areas of vagueness in the introduction and results. Figures could have minor alterations to improve their clarity. The article presents an interesting and innovative approach to delineating salt marsh edge.

Comments

The main issue with the study is a lack of an accuracy assessment. If possible, in situ marsh scarps should be collected to compare with the methods presented here. However, if that can’t be done, an accuracy assessment should still be conducted. The assessment could identify the most accurate method i.e., SfM derived UAS MEED approach and compare the other methods to the results of that method. Alternatively, a digitized shoreline using the full suite of data could be used for an accuracy assessment. However, comparing the various outcomes is not enough on its own.

A discussion of lidar within the salt marsh is warranted, including topobathymetric lidar and methods of DEM creation in salt marsh environments e.g., minimum elevation binning has been suggested as more accurate [1].

The paper's introduction lacks a clear and compelling reason for the MEIP method opposed to the many other salt marsh classification approaches. Especially given that this metric is focused on one salt marsh change metric and doesn’t include interior change at all, e.g., lines 34-35 “it is imperative to quantify not only marsh elevation, but also marsh areal coverage.” However, the research doesn’t present a method for measuring areal coverage of the salt marsh. Perhaps additional research that suggests edge erosion and collapse from eutrophication or herbivory as significant drivers of loss would improve the introduction [2-3].

There should be additional clarity as to why the MEIP method is included, e.g., is MEIP merely a comparison method? Previous studies that explicitly found the salt marsh edge could be beneficial to the introduction. A few studies of possible interest include waterline mapping to derive a DEM, salt marsh edge change analysis with NAIP, and Landsat to determine shorelines [4-6].

Minor Comments:

Figures: List the UTM zone of the insets to make it clear to users what the units are. Remove decimal place precision from the grids- this clutter the labels. Redefine scale bar divisions to be discernible (Figure 4-8).

Line 161: What were the slightly different elevations or smoothing lengths? Why were these chosen, and how was the improvement in the results quantified? Description of parameters and assessment need to be clarified.

Line 172 – Not clear what the “unwanted parts of the marsh” are. This phrase needs to be defined and clarified. Are these channel and ditch mouths? If so, explain why these are undesirable e.g., is discerning shoreline position beyond the capabilities of the data or merely outside the scope of the analysis?

Line 200: If you utilize the ArcGIS basemap list an acquisition date for the imagery by using this webamp: https://www.arcgis.com/home/webmap/viewer.html?webmap=c03a526d94704bfb839445e80de95495 webmap.

Line 353: Clusters are being assigned classes during the “reclassification” step. The term doesn’t adequately define the process.

Line 375: Replace “founding” with “funding” 

References

[1] Schmid, K.A., Hadley, B.C. and Wijekoon, N., 2011. Vertical accuracy and use of topographic LIDAR data in coastal marshes. Journal of Coastal Research27(6A), pp.116-132.

[2] Holdredge, C., Bertness, M.D. and Altieri, A.H., 2009. Role of crab herbivory in dieoff of New England salt marshes. Conservation Biology23(3), pp.672-679.

[3] Wigand, C., Roman, C.T., Davey, E., Stolt, M., Johnson, R., Hanson, A., Watson, E.B., Moran, S.B., Cahoon, D.R., Lynch, J.C. and Rafferty, P., 2014. Below the disappearing marshes of an urban estuary: historic nitrogen trends and soil structure. Ecological Applications24(4), pp.633-649.

[4] White, S.M. and Madsen, E.A., 2016. Tracking tidal inundation in a coastal salt marsh with Helikite airphotos: Influence of hydrology on ecological zonation at Crab Haul Creek, South Carolina. Remote Sensing of Environment184, pp.605-614.

[5] Campbell, A. and Wang, Y., 2019. High Spatial Resolution Remote Sensing for Salt Marsh Mapping and Change Analysis at Fire Island National Seashore. Remote Sensing11(9), p.1107.

[6] Kuleli, T., Guneroglu, A., Karsli, F. and Dihkan, M., 2011. Automatic detection of shoreline change on coastal Ramsar wetlands of Turkey. Ocean Engineering38(10), pp.1141-1149.

Round 2

Reviewer 2 Report

The authors have adequately addressed my review comments, and I am recommending that the paper be accepted for publication. One wording edit: on line 61: I would change “submeter scale” to “submeter GSD.”

Reviewer 3 Report

The article is markedly improved, and I have no additional concerns.

Minor:

Line 162 – Change “This data was” to “These data were”